

# Importance of active layer freeze-thaw cycles on the riverine dissolved carbon export on the Qinghai-Tibet Plateau permafrost region

Chunlin Song[1,2], Genxu Wang[1], Tianxu Mao[3], Xiaopeng Chen[4], Kewei Huang[1,2], Xiangyang Sun[1] and Zhaoyong Hu[1]

[1] Institute of Mountain Hazards and Environment, Chinese Academy of Sciences, Chengdu, China
[2] University of Chinese Academy of Sciences, Beijing, China
[3] Guizhou University, Guiyang, China
[4] Shanxi Agricultural University, Jinzhong, China

Corresponding author
Genxu Wang, wanggx@imde.ac.cn

## ABSTRACT

The Qinghai-Tibet Plateau (QTP) is experiencing severe permafrost degradation, which can affect the hydrological and biogeochemical processes. Yet how the permafrost change affects riverine carbon export remains uncertain. Here, we investigated the seasonal variations of dissolved inorganic and organic carbon (DIC and DOC) during flow seasons in a watershed located in the central QTP permafrost region. The results showed that riverine DIC concentrations ($27.81 \pm 9.75$ mg L$^{-1}$) were much higher than DOC concentrations ($6.57 \pm 2.24$ mg L$^{-1}$). DIC and DOC fluxes were 3.95 and 0.94 g C m$^{-2}$ year$^{-1}$, respectively. DIC concentrations increased from initial thaw (May) to freeze period (October), while DOC concentrations remained relatively steady. Daily dissolved carbon concentrations were more closely correlated with baseflow than that with total runoff. Spatially, average DIC and DOC concentrations were positively correlated with vegetation coverage but negatively correlated with bare land coverage. DIC concentrations increased with the thawed and frozen depths due to increased soil interflow, more thaw-released carbon, more groundwater contribution, and possibly more carbonate weathering by soil $CO_2$ formed carbonic acid. The DIC and DOC fluxes increased with thawed depth and decreased with frozen layer thickness. The seasonality of riverine dissolved carbon export was highly dependent on active layer thawing and freezing processes, which highlights the importance of changing permafrost for riverine carbon export. Future warming in the QTP permafrost region may alter the quantity and mechanisms of riverine carbon export.

## INTRODUCTION

Understanding the role of permafrost in the global carbon cycle is becoming increasingly important since climate warming and permafrost degradation can significantly mobilize the permafrost carbon pool. Permafrost carbon stock is more than twice the size of the

entire atmospheric carbon pool, which becomes subject to enhanced biogeochemical cycling upon permafrost warming and degradation (*Schuur et al., 2008*, *2015*; *Zimov, Schuur & Chapin, 2006*). Permafrost soil can release and process carbon from land to aquatic system or atmosphere when permafrost degrades (*Frey & McClelland, 2009*; *Schuur et al., 2008*). Thawing permafrost and deepening active layer (the soil layer above the permafrost that thaws during the summer and refreezes during the winter) can strongly affect the catchment hydrology due to increased soil filtration, deeper flow path, increased retention time, and changed soil adsorptive capacity (*Frey & McClelland, 2009*; *Kawahigashi et al., 2004*; *Walvoord & Kurylyk, 2016*; *Wang, Hu & Li, 2009*). These response of hydrological changes to permafrost degradation are also expected to change the patterns of riverine carbon export (*Frey & McClelland, 2009*), which is not only a major component of carbon cycle (*Cole et al., 2007*), but also an indicator of the responses of adjacent terrestrial ecosystems to environmental change (*Spencer et al., 2015*).

Riverine carbon export is closely coupled with watershed hydrology (*Hartmann, 2009*; *McClelland et al., 2016*; *Raymond et al., 2007*; *Raymond & Oh, 2007*; *Vihermaa et al., 2016*). Other factors including soil carbon, vegetation, climate change, and human activities were also considered to be important regulators on riverine carbon export (*Hope, Billett & Cresser, 1994*; *Lepistö, Futter & Kortelainen, 2014*; *Raymond et al., 2007*, *2008*). Permafrost distribution and change have important impacts on carbon production, cycling, and transport by rivers and streams (*Frey & McClelland, 2009*; *Olefeldt & Roulet, 2014*; *Tank et al., 2012a*; *Vonk et al., 2015a*). In permafrost regions, the lateral carbon loss that accompanies water flow in permafrost regions is highly sensitive to permafrost conditions (*Frey & McClelland, 2009*). The deepening active layer could increase dissolved organic carbon (DOC) concentration and flux since more available organic material (*Petrone et al., 2006*; *Raymond et al., 2007*). Increased pathway and longer travel time due to active layer deepening could have important effects on dissolved carbon process and transport (*Frampton & Destouni, 2015*). Thawing of an active layer could result in the decrease of DOC and increase of dissolved inorganic carbon (DIC) due to DOC mineralization and respiration (*Drake et al., 2015*; *Striegl et al., 2005*, *2007*). Subsurface flow pathway and groundwater change due to deepening active layer may result in riverine carbon dynamics (*Walvoord & Striegl, 2007*). While these studies provided valuable information to understand the permafrost role on riverine carbon export, knowledge about how river dissolved carbon export is affected by the seasonal change of an active layer in continuous permafrost is lacking. Thaw and freeze depth change of an active layer are associated with the change of subsurface water storage, flow path, and residence time (*Tetzlaff et al., 2015*; *Walvoord & Kurylyk, 2016*). The seasonal change of the soil active layer thaw depth can expose organic layers and mineral components of the soil profile to subsurface flow in different magnitude, which are expected to affect the riverine carbon export.

As the largest low-middle latitude permafrost region in the world, the Qinghai-Tibet Plateau (QTP) is covered by a permafrost area of 1,060,000 km$^2$ (*Zou et al., 2017*). The permafrost of the QTP highland is warmer than Arctic permafrost and is considered to be more sensitive to climatic warming than high-latitude permafrost (*Cheng & Wu, 2007*). The QTP permafrost stores approximately 12.72 Pg C of soil organic carbon (SOC)

(*Zhao et al., 2018*) and 15.19 Pg C of soil inorganic carbon (SIC) in top one meter soil (*Yang et al., 2010a*), which has the potential to be quickly released from the deepening active layer to fluvial systems under climate warming scenarios (*Frey, 2005*; *Harden et al., 2012*). Many permafrost riverine carbon studies focused on the boreal and Arctic region, but the warmer QTP permafrost has been significantly underrepresented due to a dearth of data. A previous study at northern QTP found that DOC concentration decreased as thaw depth increased (*Mu et al., 2017*). While it is unknown about the DIC exports changes with the freeze-thaw cycles of an active layer, it has been documented that active layer freeze-thaw cycles can significantly affect the seasonal runoff components and runoff generation processes (*Wang, Hu & Li, 2009*, *Wang et al., 2017*). We hypothesized that the quantity of riverine carbon export might be ultimately affected by the active layer seasonal thaw and freeze processes for permafrost underlain rivers. To address these knowledge gaps and test the hypothesis, we conducted field investigations in a permafrost watershed of the QTP in 2014 and 2016 non-frozen seasons (May–October). A particular emphasis of our study was on the dissolved carbon export in thawing and freezing periods. In this study, we (1) characterized the spatiotemporal variations of riverine DIC and DOC concentrations, fluxes and the influencing factors, and (2) elucidated the impacts of active layer freeze-thaw cycles on riverine carbon export in the permafrost basin of QTP.

## METHODS

### Study area

This study was conducted in five catchments located in the Fenghuo Mountain region of the QTP, which is covered by continuous permafrost (Fig. 1). Table 1 shows the characteristics of the catchments in detail. Catchment 2 to 5 were nested in Catchment 1. The five catchments constitute an entire watershed, Zuomaokong watershed, which is a tributary of the Tongtian River that represents the headwaters of the Yangtze River. The total watershed area is 117.0 km$^2$, and the elevation of the watershed ranges from 4,720 to 5,392 m above sea level.

The study area is characterized by a cold and dry continental alpine climate. The mean annual air temperature is −5.2 °C, and the mean annual precipitation is 328.9 mm. The highest temperature (averagely 11.6 °C) in each year occurs in July or August, and the monthly mean air temperature is above 0 °C from May to September. The relative humidity of the air ranges from 17% to 96% in the winter and summer, respectively. Precipitation events that occur from June to September account for 85% of the total annual rainfall amount, and peak precipitation occurs in July or August. During the freezing season from November to the following April, the total precipitation (snowfall) is usually less than 21 mm (*Song et al., 2017*). The snow cover during the winter season is sporadic and thin, thus, snowmelt events rarely occur during the spring thaw period (*Wang et al., 2017*).

Permafrost depths of the study area vary from 80 to 120 m, and active layer depths range from 1.3 to 2.5 m (*Wu & Liu, 2004*). The soil types in the study area are primarily classified as Mattic-Gelic Cambisols (alpine meadow soil) (*Wang, Hu & Li, 2009*). The vegetation within the study area is dominated by alpine meadow (70% in our study watershed) and alpine swamp meadow, which are the most widespread types of

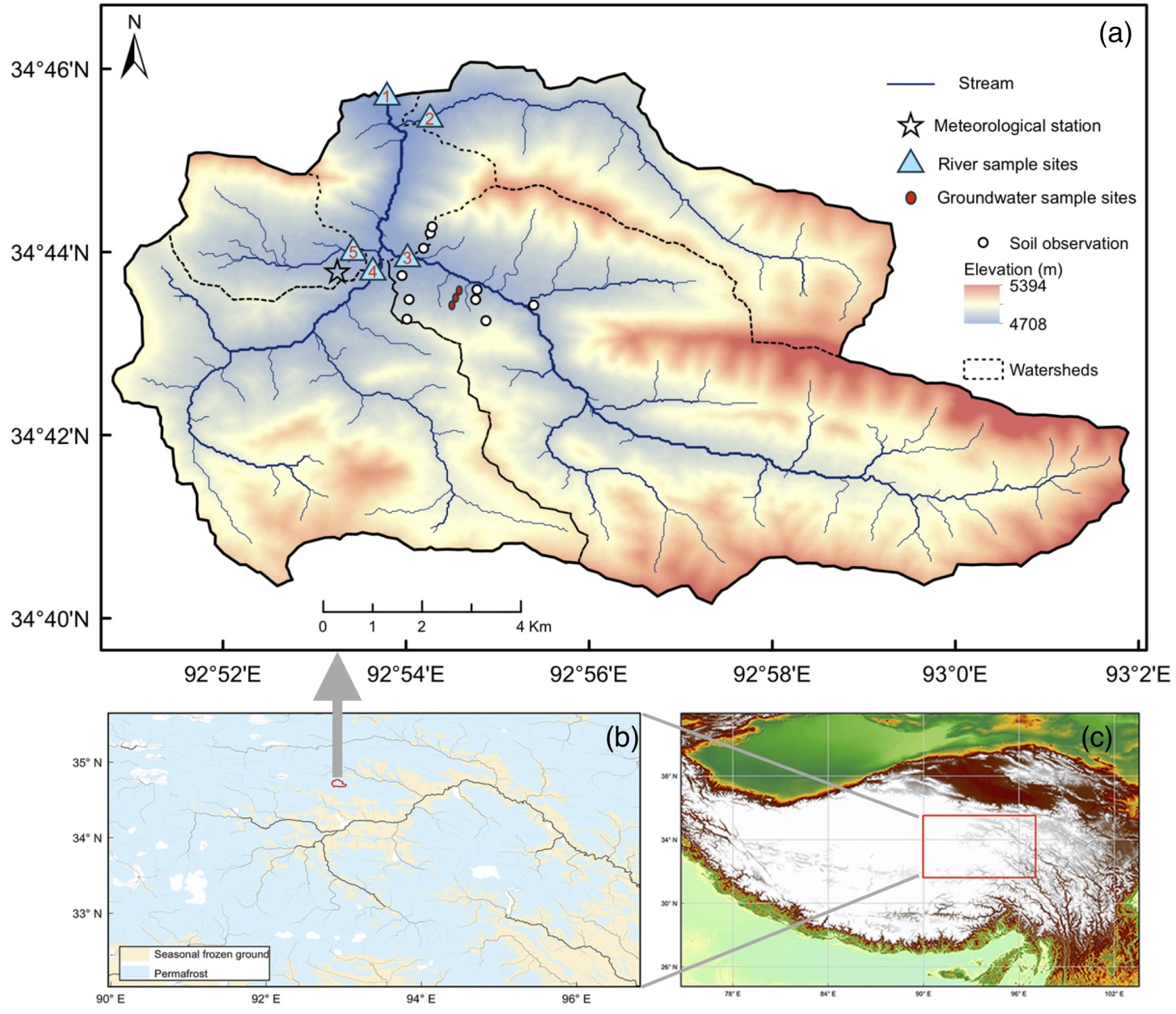

**Figure 1 Map of the study area.** Map of the study area. (A) shows the study watershed, while the dashed lines are boundaries of the sub-catchments. Sample sites and observation sites were marked on the map. (B) shows the distribution of permafrost in the Qinghai-Tibet Plateau (*Zou et al., 2017*). (C) shows the location of the Qinghai-Tibet plateau. Figure parts were generated by ArcGIS Desktop 10.4 software (http://www.esri.com/software/arcgis).

vegetation across the QTP permafrost region. The dominant plant species are sedges *Kobresia pygmaea* and *K. humilis* (Fig. S1). Catchment 5 has the highest vegetation coverage (Vc) (57.3%), whereas Catchment 2 has the lowest Vc (32.3%). Livestock grazing pressure in the study area is light.

The frozen season usually begins in October and ends in the following April. During the frozen season, the river channel contains no open liquid water for sampling. The annual

**Table 1 Characteristics of the sub-catchments of the watershed.**

| Site | Sampling sites coordinates | | Area (km²) | Mean elevation (m) | Elevation ≥5,000 m (%) | Vegetation coverage (%) | Wetland cover (%) | Bare land cover (%) |
|---|---|---|---|---|---|---|---|---|
| | Latitude | Longitude | | | | | | |
| Catchment 1 | 34°45′48″N | 92°53′49″E | 117.0 | 4,922 | 22.2 | 43.4 | 4.5 | 47 |
| Catchment 2 | 34°45′18″N | 92°53′59″E | 18.3 | 4,915 | 20.2 | 32.3 | 3.3 | 55 |
| Catchment 3 | 34°44′8″N | 92°53′45″E | 56.4 | 4,948 | 32.1 | 37.9 | 1.9 | 52 |
| Catchment 4 | 34°44′2″N | 92°53′43″E | 30.1 | 4,905 | 10.5 | 52.5 | 2.4 | 39 |
| Catchment 5 | 34°44′2″N | 92°53′42″E | 6.8 | 4,834 | 5.7 | 57.3 | 10.7 | 32 |

Note:
Basin area and topography were obtained from ArcGIS 10.2 with GMTED2010 DEM data. Land cover of the basin were derived from SPOT-7 Satellite Images.

runoff processes out of the frozen season have two flooding periods and two dry periods. These periods include a summer flooding period that starts in July followed by a slight recession in early August, an autumn flooding period in September with baseflow recession from late September to October (*Wang, Liu & Liu, 2011*).

## Field observations

At the outlet of the entire watershed (Catchment 1), flow depths were recorded continuously using a U20 HOBO Water Level Logger (MicroDAQ, Contoocook, NH, USA) at a 1-h interval (0.021 m resolution). The stream velocity was measured using an FP101 Digital Water Velocity Meter (GLOBAL WATER, College Station, TX, USA, 0.03 m s$^{-1}$ resolution). We used the widely used velocity-area method to calculate river discharge in this study. Soil temperature and moisture of 10 sites at depths of 0.05, 0.2, 0.5, 1, and 1.6 m were measured using temperature and moisture sensors (5TM; Decagon Devices, Pullman, WA, USA) installed inside the Catchment 3 (Fig. 1), and the measurements were recorded by data loggers (EM50; Decagon Devices, Pullman, WA, USA). Soil moisture and temperature were measured simultaneously at 30-min intervals. The daily average soil moisture and temperature data were processed in R (version 3.3.2), using the "xts" package. The thawed depth during the thawing seasons and the frozen depth during the frozen seasons of the soil active layer (Figs. S2 and S3) were determined based on the daily soil temperature (*Frauenfeld et al., 2004*). We used daily soil temperature measurements at different depths to interpolate the depth of 0 °C isotherm depth. Then the 0 °C isotherm depths were used to determine thaw and freeze depth. Daily mean soil temperature and moisture were displayed in Fig. S4. Meteorological data, including precipitation and air temperature, were collected at the meteorological station located in the Zuomaokong watershed.

## Sample collection and analyses

We collected stream water, rainwater, and groundwater samples for DOC and DIC analyses. Stream water samples were collected at the outlet of the five catchments from May to October in 2014 and 2016. We sampled about weekly in 2014 and daily in 2016 thaw and freeze seasons. Sampling work was conducted in the middle of the day. Stream water samples were collected at approximately half depth from the water surface to the river bed and center of the transects at the outlets of the sub-catchments (Fig. 1). Rainwater

**Table 2 Coefficients (±SD) and $R^2$ values for LOADEST model 9 (ln(flux) = $a_0 + a_1$lnQ + $a_2$lnQ$^2$ + $a_3$Sin(2πdtime) + $a_4$Cos(2πdtime) + $a_5$dtime + $a_6$dtime$^2$).**

|  | $R^2$ | $a_0$ | $a_1$ | $a_2$ | $a_3$ | $a_4$ | $a_5$ | $a_6$ |
|---|---|---|---|---|---|---|---|---|
| DIC | 0.9836 | 7.4367 ± 0.0756 | 0.942 ± 0.0275 | 0.0289 ± 0.0137 | 0.3108 ± 0.0256 | 0.1856 ± 0.0813 | 0.2869 ± 0.0185 | −0.4002 ± 0.055 |
| DOC | 0.9794 | 5.8034 ± 0.0805 | 1.0102 ± 0.0293 | 0.047 ± 0.0146 | 0.0749 ± 0.0273 | 0.049 ± 0.0866 | 0.3158 ± 0.0197 | −0.1727 ± 0.0585 |

**Note:**
Flux in kg/day, Q in ft$^3$/s, lnQ = ln(streamflow)—center of ln(streamflow), dtime = decimal time—center of decimal time.

samples were collected near the meteorological station after each day of rain and was assumed to represent the entire watershed precipitation. Groundwater samples were collected in three groundwater observation wells (~2 m depth below ground, Fig. 1) inside the watershed at a 3-day interval in 2016. Fresh stream water samples were filtered by vacuum filtration method immediately after sampling through 0.45 μm pore size glass fiber filters (Shanghai Xinya, Shanghai, China). After filtration, 100 mL of each water sample were poured into pre-rinsed high-density polyethylene bottles with tight-fitting caps and then immediately stored under frozen and dark conditions. The samples were kept frozen before the analysis. We used an Elementar Vario TOC Select Analyzer (Elementar, Langenselbold, Germany) to analyze the DOC, DIC, and total dissolved nitrogen (TDN) concentrations of the water samples within one week after sampling. Results of triple injections of indicated the analytical precision <3%. Base cations ($Na^+$, $Mg^{2+}$, $K^+$, $Ca^{2+}$) from river water samples during 2014 were analyzed with ICS-90 Ion Chromatography System (Dionex, Sunnyvale, CA, USA).

## Calculations and statistics

We combined dissolved carbon measurements and continuous discharge data and used the USGS load estimator (LOADEST) program (*Runkel, Crawford & Cohn, 2004*) to derive fluxes of DIC and DOC of our study watershed. LOADEST uses daily flow and carbon concentration data to establish relationships between flow and concentration and applies these relationships to the complete daily discharge record and then derive flux estimates. LOADEST centers the discharge and dissolved carbon concentration data to eliminate multicollinearity and selects one of nine predefined regression models to fit the data, based on the Akaike Information Criterion and Schwarz Posterior Probability Criteria. This method has been widely used in the Arctic river flux calculation (*Tank et al., 2012b*; *Raymond et al., 2007*). In our study, the model 9 was selected by the LOADEST for DIC and DOC flux estimation:

$$\ln(\text{flux}) = a_0 + a_1 \ln Q + a_2 \ln Q^2 + a_3 \text{Sin}(2\pi \text{dtime}) + a_4 \text{Cos}(2\pi \text{dtime}) + a_5 \text{dtime} + a_6 \text{dtime}^2 \tag{1}$$

where flux is provided in kg/day, Q is river discharge (ft$^3$/s), lnQ equals ln(streamflow) minus center of ln(streamflow), dtime equals decimal time minus center of decimal time, other parameters (i.e., $a_0$, $a_1$, $a_2$, $a_3$, $a_4$, $a_5$, $a_6$) are shown in Table 2. LOADEST produces daily, monthly and annually flux values using maximum likelihood estimation, adjusted maximum likelihood estimation (AMLE), and least absolute deviation statistical

approaches. All three output types provided similar results, the AMLE results were used in this study. LOADEST models explained large variability of DIC and DOC, with $R$-squared values of 0.9836 and 0.9794 for DIC and DOC, respectively.

Baseflow is usually related to subsurface flow and groundwater flow. It is necessary to test the relations between baseflow and riverine carbon since permafrost degradation could enhance soil drainage and recharge, suprapermafrost flow, groundwater-surface water exchange, sub-permafrost flow, and baseflow (*Walvoord & Kurylyk, 2016*). The full observation of daily river runoff during our study period allows us to use the digital filter method (*Lyne & Hollick, 1979*) for baseflow separation. To be specific, we generated daily baseflow data in 2014 and 2016 of Catchment 1 with a web-based hydrograph analysis tool, which was built on the BFLOW and Eckhardt filter modules (*Lim et al., 2005*).

Pearson's correlation analysis was performed to determine potential relationships between hydrological and surface meteorological factors and dissolved carbon concentrations. The correlation result was illustrated as a correlation matrix, which displayed Pearson's correlation coefficients ($r_p$) with significance levels and scaled kernel regression smoothing lines. Linear correlation analyses were performed to assess the impact of spatial difference and thaw/freeze depths on DIC and DOC concentrations. Since Catchment 3 has the most intense observations of active layer thaw, we use the data from Catchment 3 to analysis the impact of freeze and thaw depth on dissolved carbon export.

## RESULTS

### Temporal and spatial variations of dissolved carbon

The DIC and DOC concentrations at the outlet of the Zuomaokong watershed displayed different fluctuation patterns from May to October (Figs. 2A and 2B). The DIC concentrations showed larger fluctuations than those of DOC from May to October in both 2014 and 2016. The coefficients of variation (CVs) of DIC in 2014 and 2016 were 0.39 and 0.19, while the CVs of DOC in 2014 and 2016 were 0.22 and 0.18, respectively. The DIC concentrations ranged from 8.15 to 30.9 mg L$^{-1}$ and 21.58 to 47.5 mg L$^{-1}$ in 2014 and 2016, respectively. The DOC concentrations during 2014 and 2016 ranged from 2.33 to 6.38 mg L$^{-1}$ and 4.99 to 13.79 mg L$^{-1}$, respectively. Large variations of dissolved carbon concentrations appeared both seasonally and annually. Generally, the DIC concentrations increased with Julian day during the open-water seasons (Fig. 2, slopes of trend lines are 0.11 and 0.09 for 2014 and 2016), while the DOC concentrations were relatively steady. Similar patterns were also found in other four sub-catchments (Fig. S6).

Seasonal variations of the dissolved carbon fluxes from the entire watershed were high in 2014 and 2016 (Fig. 3). The highest DIC and DOC fluxes (DICF and DOCF) occurred in September, whereas the lowest DICF and DOCF occurred in May. Not surprisingly, the DICF and DOCF were closely related to river discharge since fluxes were calculated as the function of concentration and discharge. The largest runoff depths and carbon fluxes appeared simultaneously in September. However, the month with the highest precipitation did not have the highest carbon flux. The fluvial DIC fluxes were 2.76 g m$^{-2}$ year$^{-1}$ in 2014
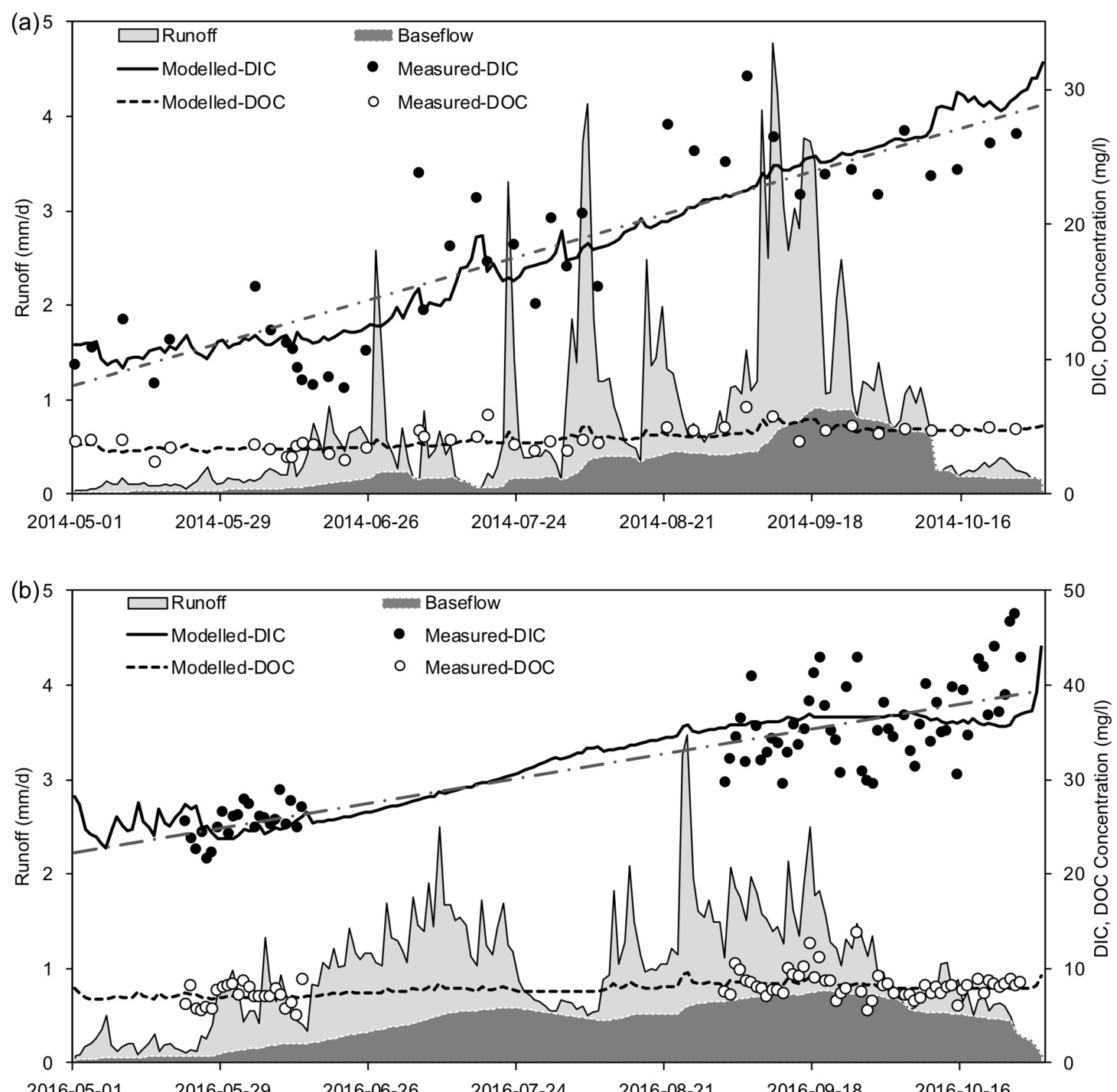

**Figure 2 Runoff processes and time series of the riverine exported DIC and DOC concentrations at the outlet of Zuomaokong watershed (Catchment 1) during 2014 (A) and 2016 (B) freeze-free periods (May–October).** The round solid and circle points are directly observed data and the solid lines are calculated from LOADEST model results. Daily discharge and baseflow are displayed with gray shades. The dashed straight lines are linear trends of measured DIC concentrations, while no significant trends were found for DOC concentrations. Note the linear trend lines based on measured DIC: (A), $y = 0.11x - 5.71$, $R^2 = 0.72$; (B), $y = 0.09x + 10.76$, $R^2 = 0.71$; $y$ represents DIC concentrations, $x$ represents the day of the year.

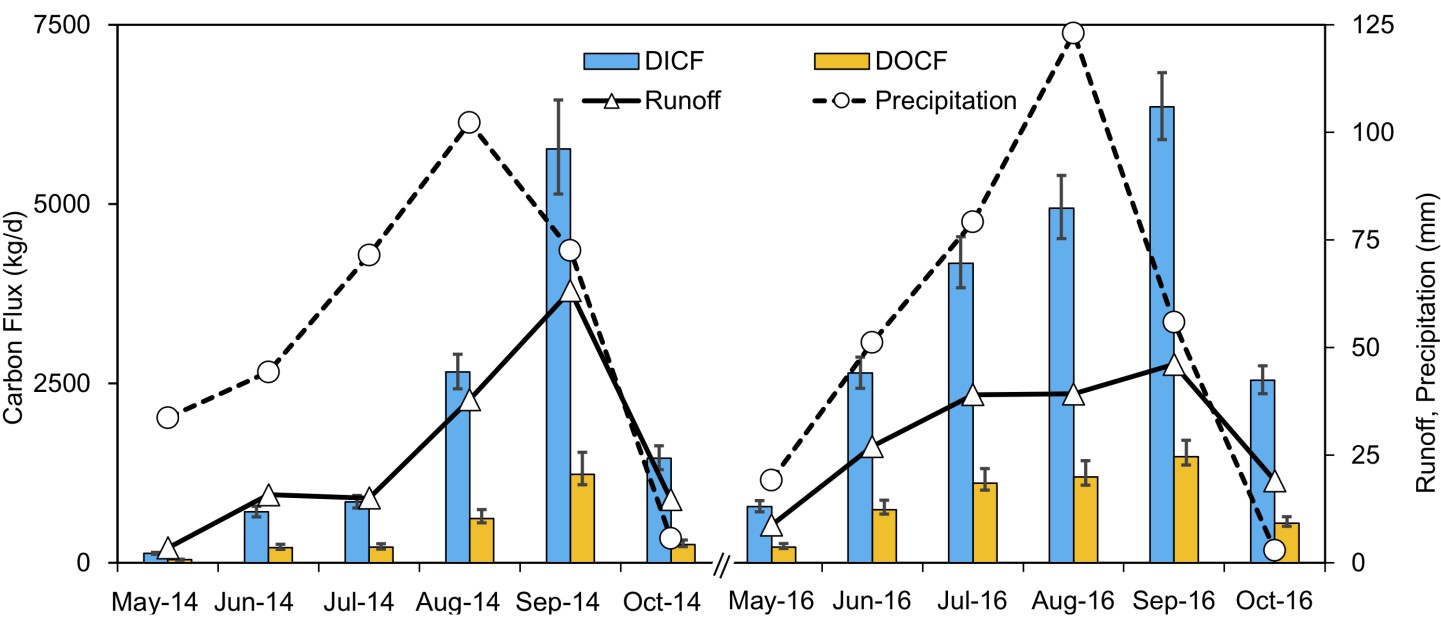

**Figure 3 Monthly average DIC and DOC fluxes in kg/day, monthly total discharge, and precipitation variations of the entire watershed (Catchment 1) during 2014 and 2016.** The error bars indicate the upper and lower 95% confidence intervals of LOADEST results.

and 5.14 g m$^{-2}$ year$^{-1}$ in 2016, whereas the DOC fluxes were 0.61 g m$^{-2}$ year$^{-1}$ in 2014 and 1.27 g m$^{-2}$ year$^{-1}$ in 2016. The average riverine DIC and DOC fluxes were 3.95 and 0.94 g m$^{-2}$ year$^{-1}$, respectively.

The average DIC concentration was markedly higher than the average DOC concentration at the watershed outlet and in all of the sub-catchments (Table 3), which demonstrated that DIC was the dominant form of dissolved carbon export in QTP permafrost rivers. In the meantime, dissolved carbon concentrations showed spatial differences among the sub-catchments. The mean DIC and DOC concentrations of the entire watershed measured at the outlet (Catchment 1) were 27.81 ± 9.75 and 6.57 ± 2.24 mg L$^{-1}$, respectively. Catchment 4 had the highest mean DIC and DOC concentrations, whereas Catchment 3 displayed the lowest DIC concentration, and Catchment 2 displayed the lowest DOC concentration (Table 3).

Additionally, the groundwater samples had DIC and DOC concentrations that were nearly double those of the stream water, which were 59.28 ± 16.58 and 11.59 ± 2.53 mg L$^{-1}$, respectively. But DIC and DOC concentrations of the rainwater were much lower (3.08 ± 1.93 and 2.66 ± 1.93 mg L$^{-1}$ for DIC and DOC, respectively) than stream water.

## Factors influencing DIC and DOC: from time to space

The correlations between the time series of hydrological and meteorological variables and DIC and DOC concentrations in Catchment 1 showed that river runoff was slightly correlated with the DIC and DOC concentrations, with $r_p$ of 0.30 and 0.32, respectively. Interestingly, DIC and DOC concentrations showed closer correlations with the baseflow ($r_p$ = 0.67 and 0.54, respectively) than total river runoff. Precipitation and air temperature

**Table 3 Dissolved carbon concentrations (mg C L$^{-1}$) of stream water, precipitation, and groundwater in 2014 and 2016 freeze-free seasons.**

| Sampling location | Variables | N | Mean (±SD) |
|---|---|---|---|
| Catchment 1 | DIC | 117 | 27.81 (±9.75) |
|  | DOC | 117 | 6.57 (±2.24) |
| Catchment 2 | DIC | 111 | 27.14 (±7.60) |
|  | DOC | 111 | 5.92 (±1.76) |
| Catchment 3 | DIC | 117 | 26.92 (±9.80) |
|  | DOC | 117 | 6.36 (±2.17) |
| Catchment 4 | DIC | 109 | 31.98 (±9.90) |
|  | DOC | 109 | 9.34 (±5.97) |
| Catchment 5 | DIC | 111 | 29.89 (±11.42) |
|  | DOC | 111 | 7.91 (±2.94) |
| Precipitation | DIC | 52 | 3.08 (±1.93) |
|  | DOC | 52 | 2.66 (±1.93) |
| Groundwater | DIC | 56 | 59.28 (±16.58) |
|  | DOC | 56 | 11.59 (±2.53) |

**Note:**
DIC concentration is significantly higher than DOC for all the sampling sites. Sampling locations are marked in Fig. 1A. N column is the sample size.

showed weak correlations with DIC and DOC concentrations. Meanwhile, the DIC and DOC concentrations were closely correlated to each other, with an $r_p$ of 0.84 (Fig. 4), indicating close associations between DIC and DOC.

The different characteristics of the sub-catchments allow us to analyze the potential factors affecting the spatial variations of dissolved carbon concentrations with correlation analyses. The results showed that the dissolved carbon concentrations were closely associated with the land cover. DIC and DOC concentrations were positively correlated with Vc ($r_p$ = 0.82 and 0.91, respectively), but weakly correlated with wetland cover ($r_p$ = 0.21 and 0.32, respectively) within the five catchments (Fig. 5). The bare land coverage showed negative relationships with DIC and DOC ($r_p$ = −0.88 and −0.78, respectively). The dissolved carbon concentrations decreased as mean elevation increased since vegetation cover decreased and bare land cover increased in higher elevation catchments. Catchment size showed a limited impact on the dissolved carbon concentrations in our study watershed.

## Effects of active layer freeze-thaw cycles

Our results revealed that the thawing process had a significant impact on dissolved carbon concentrations and fluxes. The linear relationships between the DIC concentrations and the thawed depths during the 2016 thawing periods were statistically significant ($R^2$ = 0.289, $p$ = 0.007, Fig. 6A). As the active layer thawed at greater depths, the DIC concentrations in the river increased. The relationship between the DOC concentration and the thawed depth was weaker but also statistically significant with positive relations ($R^2$ = 0.162, $p$ = 0.05, Fig. 6C). In this period, the DIC and DOC concentration was 19.47 and 5.56 mg/L, respectively. The DIC and DOC fluxes showed strong positive

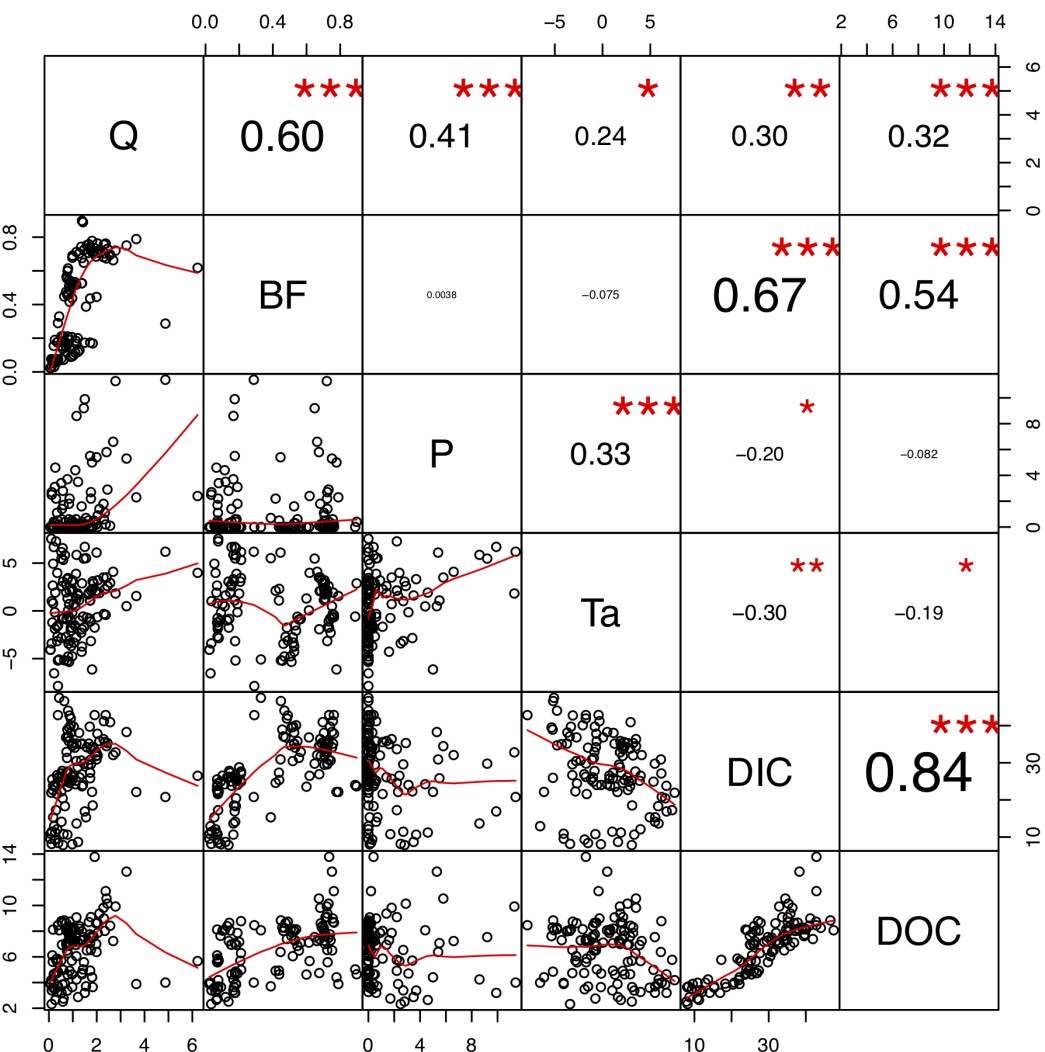

**Figure 4 Correlation matrix plot indicate relationships of dissolved carbon concentrations and hydrological and meteorological factors.** The upper panel shows positive and negative Pearson's correlation coefficients in different size (larger represent higher values). Note significance values: $^*p \leq 0.05$; $^{**}p \leq 0.01$; and $^{***}p \leq 0.001$. The red lines in lower panel are scaled kernel regression smoothers indicating potential linear and non-linear relationships. Q, discharge; BF, baseflow; P, precipitation; Ta, air temperature in °C.

relationships with the thawed depth. The DIC and DOC fluxes increased as the thawed depth increased in both 2014 and 2016 thawing seasons. The $R^2$ values and $p$-values of these relationships were in the range of 0.42–0.49 and <0.001, respectively.

As temperature decreased in October, the soil active layer started to freeze two-sided, from the surface and the bottom of the soil active layer (*Woo, 2012*). The results showed that the DIC and DOC concentrations increased as the frozen depth increased during the 2016 freezing period (Figs. 7A and 7C). In this period, the DIC and DOC concentration was 35.91 and 7.28 mg/L, respectively. As the freezing process proceeded and the frozen depth became deeper, the DIC and DOC fluxes decreased ($R^2$ = 0.5546 and 0.5147, $p < 0.001$, and $p < 0.001$, for DIC fluxes and DOC fluxes, respectively).

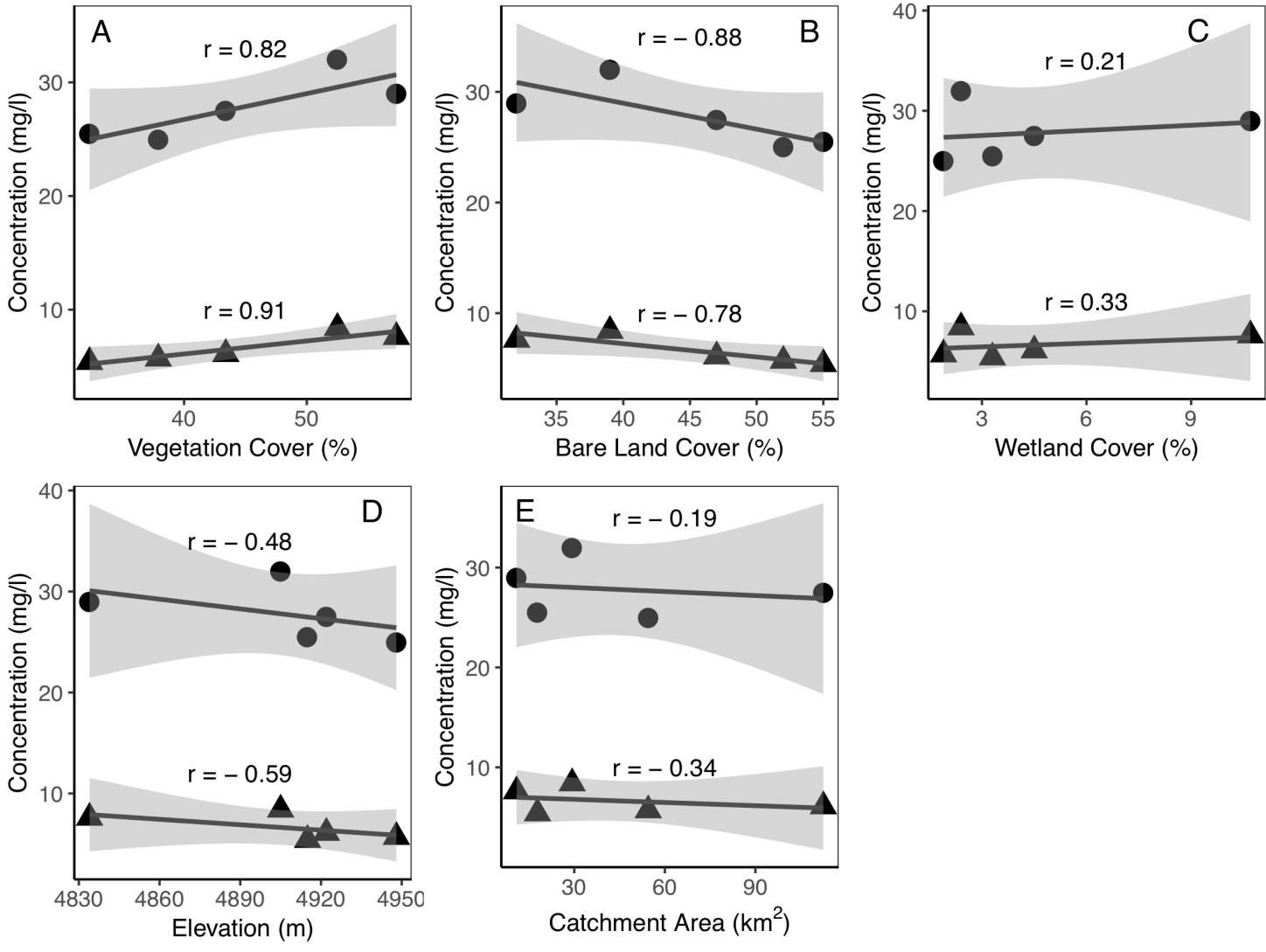

**Figure 5 Relationships between catchment characteristics and dissolved carbon concentrations of the five catchments.** (A) vegetation coverage vs. dissolved carbon concentrations; (B) bare land coverage vs. dissolved carbon concentrations; (C) wetland coverage vs. dissolved carbon concentrations; (D) mean basin elevation vs. dissolved carbon concentrations; (E) catchment area vs. dissolved carbon concentrations. The round solid dots are DIC and the triangle dots are DOC. The *r* is linear correlation coefficient. The shaded regions in the graphics are 95% confidence interval.

## DISCUSSION

### Patterns in stream water DIC and DOC

The mean DOC concentration in our study area was much lower than many comparable small Arctic streams, where these rivers showed tens to hundreds milligram per liter of DOC concentrations (*Frey, Sobczak & Mann, 2016*; *Tank et al., 2012a*; *Vonk et al., 2013*). The values of SOC density measured in Tibetan Plateau alpine meadows (9.05 kg C m$^{-2}$ in 0–100 cm depth) (*Yang et al., 2008*) were much lower than those measured in Arctic permafrost regions (32.2–69.6 kg C m$^{-2}$) (*Tarnocai et al., 2009*), which may explain the low DOC concentrations compared to most Arctic rivers. The rapid utilization of DOC after

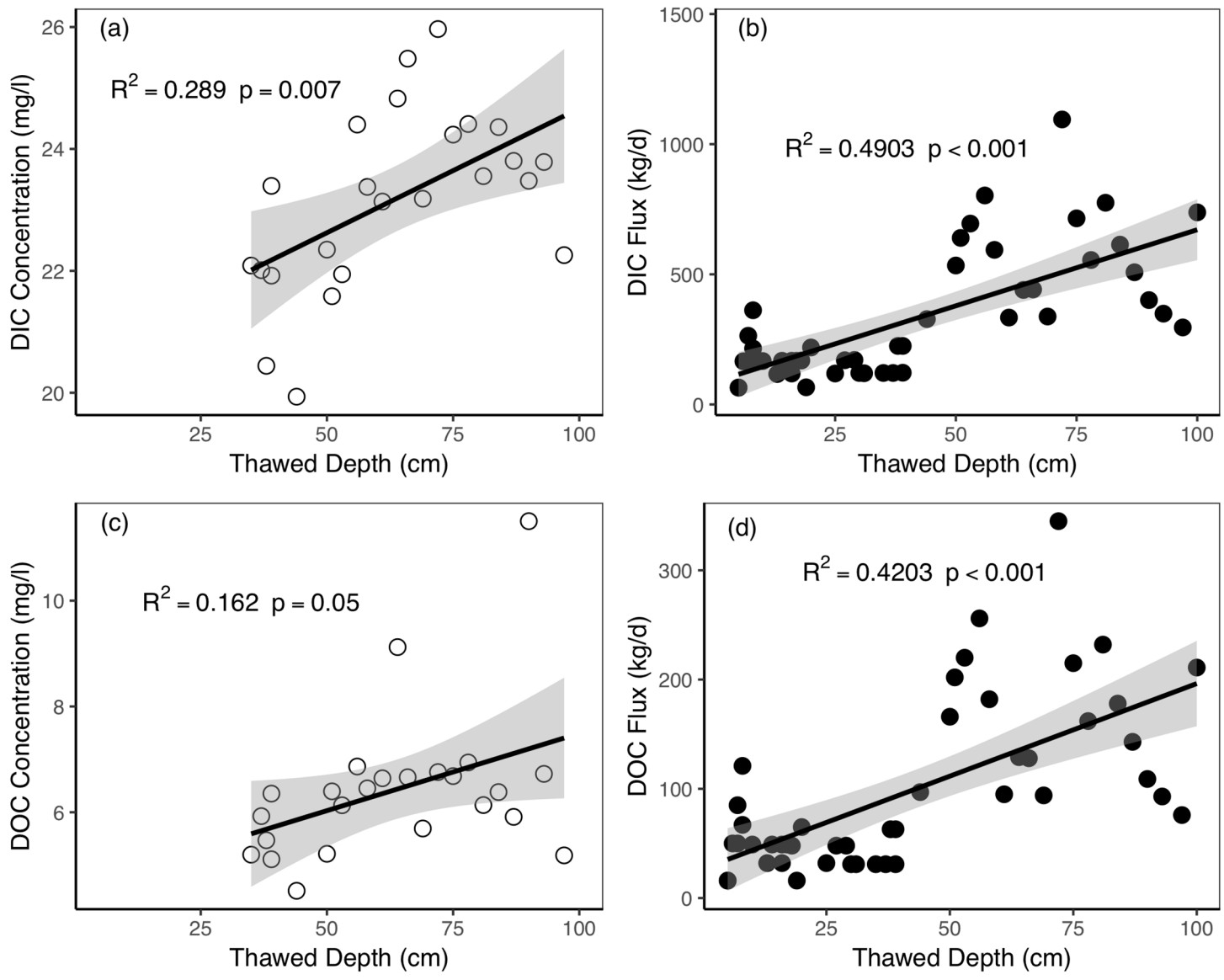

**Figure 6 Relationships between active layer thawed depths and riverine dissolved carbon concentrations and fluxes during the thawing period in Catchment 3.** (A) Thawed depth vs. DIC concentration; (B) thawed depth vs. DIC flux; (C) thawed depth vs. DOC concentration; (D) thawed depth vs. DOC flux. The shaded regions in the graphics are 95% confidence interval.

thaw (*Drake et al., 2015*; *Vonk et al., 2015a*) may also result in the low DOC in permafrost river water. The mean DIC concentrations of stream water within our field area were higher than those of Arctic permafrost rivers (*Prokushkin et al., 2011*; *Tank et al., 2012b*). One possible reason may be the high SIC density (11.87 kg C m$^{-2}$ in 0–100 cm depth) in the QTP alpine meadow (*Yang et al., 2010a*) which produce high DIC concentrations. Also, the large proportion of subsurface interflow, which is averagely 25.2% of total runoff (*Wang, Hu & Li, 2009*), could possibly introduce much soil carbon into the river channels. The dissolved carbon concentrations were higher than non-permafrost rivers (*Ran et al., 2013*; *Vihermaa et al., 2016*). The relatively high total dissolved carbon

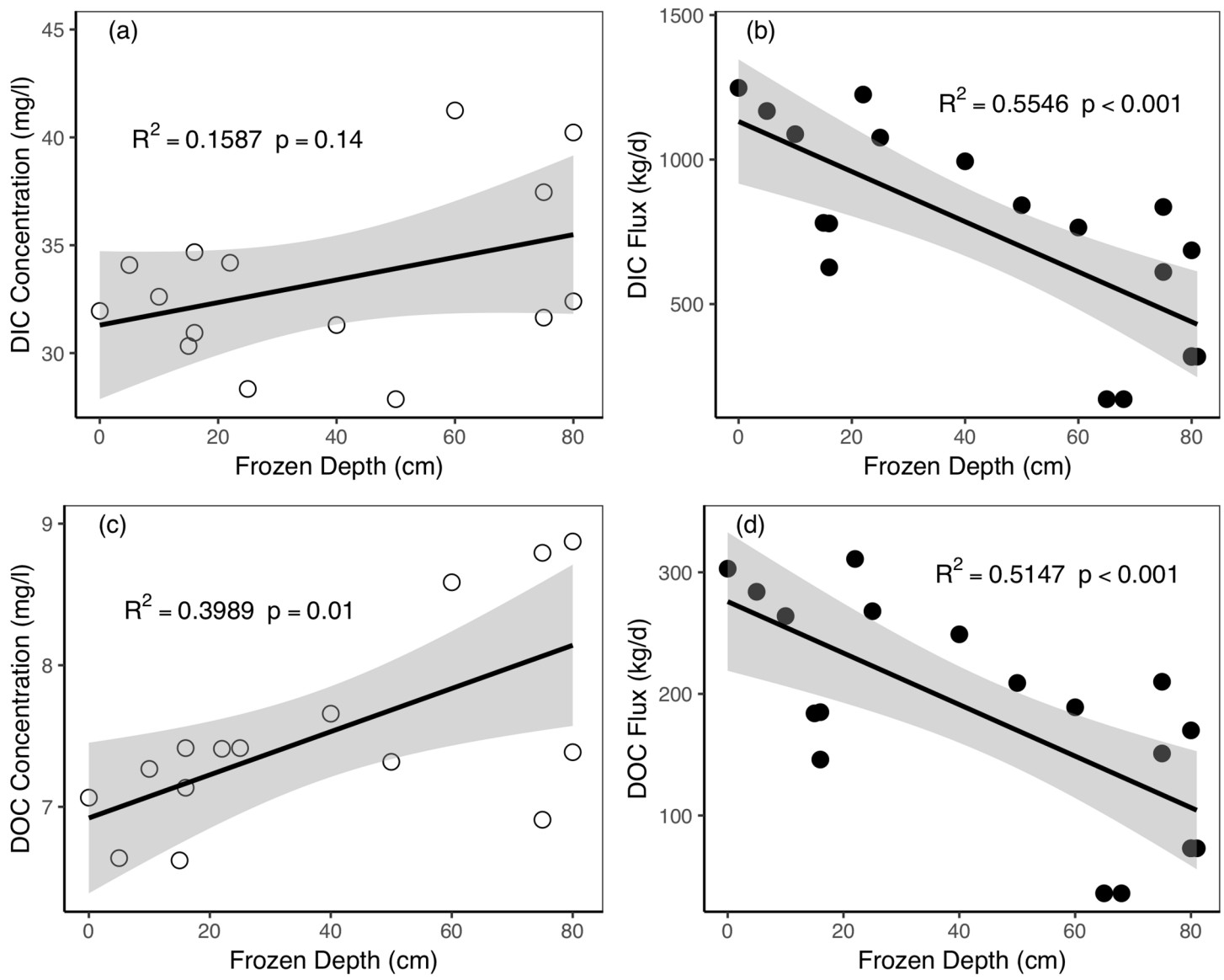

**Figure 7 Relationships between active layer frozen depths and riverine dissolved carbon concentrations and fluxes during the freezing period in Catchment 3.** (A) Frozen depth vs. DIC concentration; (B) frozen depth vs. DIC flux; (C) frozen depth vs. DOC concentration; (D) frozen depth vs. DOC flux. The shaded regions in the graphics are 95% confidence interval.

concentrations at our site indicated that the active layer in such high-elevation permafrost is a hotspot for promoting dissolved carbon loss.

Dissolved inorganic carbon was the dominant component of dissolved carbon exported to river water. The main reason for the higher DIC than DOC may be that the SIC stock is as large as 2.1 times of SOC stock in the Tibetan Plateau ecosystem (*Yang et al., 2010a*). The dominant throughflow leached inorganic material from the deep soil layer, which has higher SIC density than the upper soil layer (*Yang et al., 2010a*). According to the cation measurements of our site (Fig. S5), the dominant cations are Ca and Na+K (Na$^+$ > Ca$^{2+}$ > Mg$^{2+}$ > K$^+$), which indicate an important role of mineral weathering and evaporates
salt dissolution in the study watershed. A previous study showed widespread carbonate weathering processes ($CaCO_3 + H_2O + CO_2 \rightarrow Ca^{2+} + 2HCO_3^-$) across the Yangtze River source region (*Wu et al., 2008*), which can consume $CO_2$ and produce DIC. Besides, the average DOC and total dissolved nitrogen ratio (DOC/TDN, C/N) in our watershed were ~4.32 (Table S1), which is much lower than the global average C/N value of 22.1 (*Meybeck, 1982*). Low C/N ratio implies an abundant supply of nitrogen that could enhance the biological metabolism and DOC microbe decomposition (*Wiegner et al., 2006*). Another evidence is that a previous study at this site showed considerable amounts of old water (which is dominant in groundwater) constituted the stream flow (*Song et al., 2017*). The long retention time of old water within the basin could increase the contact of DOC to subsurface microbes, which may promote the decomposition of DOC in the active layer (*Lyon et al., 2010*; *Striegl et al., 2005*, *2012*). The higher DIC and lower DOC concentrations also indicated large proportions of subsurface flow and deeper groundwater recharge to streamflow in permafrost watersheds (*Giesler et al., 2014*; *Walvoord & Striegl, 2007*). High DIC and low DOC were also found in Arctic rivers including Tanana and Porcupine rivers (*Striegl et al., 2007*), Yenisey River (*Prokushkin et al., 2011*), and the Yukon River (*Striegl et al., 2005*) due to enhanced organic carbon mineralization.

There were increasing trends of DIC concentrations during the thawed periods of 2014 and 2016, but the DOC concentration remained relatively steady (Fig. 2). The DIC:DOC ratios increased during the thawed periods of 2014 and 2016 (Fig. S7). *Walvoord & Striegl (2007)* also found an increasing trend of the groundwater contribution and adding of DIC to streamflow in the Yukon River basin. The similar pattern of linear increase through the year in DIC was also found in Sweden rivers (*Giesler et al., 2014*). A possible explanation is that both DIC concentration and DOC concentration increased because the groundwater flow and flow path lengths increased as thawing progressed to greater depths (*Vonk et al., 2015a*), but the SIC density increased with soil layer depth (*Zhang et al., 2016*), which could result in higher DIC export when active layer thaw. The increased DIC concentration with Julian day in our study indicated deeper thawed depth and larger groundwater contribution proportion to total runoff from spring to autumn (*Wang et al., 2017*). In the autumn, the groundwater discharge contributed more than 75% of the total river runoff (*Wang et al., 2017*), which brought both high DIC concentration and DOC concentration to stream water. Future degradation of permafrost may lead to groundwater-dominated hydrologic systems among permafrost regions (*Frey & McClelland, 2009*), which will cause more carbon export through groundwater.

The correlation relationships of DIC and DOC concentration and discharge were limited compared to those of previous studies (*Li & Bush, 2015*; *Song et al., 2016*; *Tank et al., 2012b*; *Tian et al., 2015*). The dilution of dissolved carbon during high flow periods and the large proportion of groundwater recharge during low flow periods may weaken the relationship between dissolved carbon concentrations and discharge (*Tank et al., 2012b*). Baseflow was more closely related to dissolved carbon concentrations than total runoff. Baseflow contains shallow subsurface flow and groundwater flow, which could directly leach soil carbon to water channel when active layer thaw. As a result, more riverine carbon originated from the baseflow rather than the overland flow. The role of baseflow for

permafrost riverine carbon export will become increasingly important in a warming climate. As permafrost degrades, enhanced baseflow and subsurface hydrological activities (*Frey et al., 2007*; *Walvoord & Kurylyk, 2016*) could bring more permafrost carbon to aquatic systems. Our results highlight the importance of dissolved carbon supply from baseflow to river flow.

The mean river runoff in 2016 was 20% larger than 2014 (1.27 vs. 1.06 $m^3$/s of daily mean discharge). This may explain the dissolved carbon concentrations and fluxes in 2016 were larger than those of 2014. The carbon fluxes results of QTP differed from measurements taken in the Yenisey basin, where carbon export was dominated by DOC (*Prokushkin et al., 2011*). The DIC flux constituted 81% of the total dissolved carbon flux, and this proportion was close to Yukon rivers at Eagle (80%) and Pilot Station (75%) (*Striegl et al., 2007*). In 2014 and 2016, the total riverine dissolved carbon exported from the Zuomaokong watershed were 431 and 818 t C year$^{-1}$, respectively. The higher dissolved carbon flux in 2016 was due to the higher dissolved carbon concentrations and larger river runoff in 2016.

The spatial variations in the dissolved carbon concentrations of the five catchments showed both the DIC and DOC concentrations are higher in catchments with high Vc values (Fig. 5). The catchments with larger Vc will have higher ecosystem productivity and SOC density (*Wang et al., 2002*, *2008*) and thus contain more abundant sources for riverine carbon. Previous studies at this site showed that the higher the Vc is, the higher the soil temperature during the thawing and freezing period will be (*Wang et al., 2012*). High vegetation covers also led to earlier thaw-rise times (*Wang et al., 2012*). Thus, the mean dissolved carbon concentration increased because the higher soil temperatures and earlier thaw-rise times make carbon from the thawed soil available to the water flow. The coupling between vegetation and soil thermal conditions caused the differences in dissolved carbon concentrations under different vegetation cover. Conversely, catchments with larger bare land cover could have lower soil carbon density and later thaw-rise time, which decrease the dissolved carbon concentrations. Dissolved carbon concentrations decreased with the mean elevation of the catchment may be due to the high-elevation catchments have greater areas of bare land and snow cover. Wetland distribution can enhance the production of organic carbon (*Huntington & Aiken, 2013*) and have high DOC concentration in the soil profile (*Chen et al., 2017*), which explained the positive correlation between DOC and wetland coverage.

## Active layer dynamics and riverine carbon export

During the thawing period, the DIC concentration increased as the thawing processes progressed (Fig. 6). Thawed permafrost could enhance the connection between water and soil horizons and increase carbon concentrations (*Vonk et al., 2015a*). During the thawing period in spring, as air and soil temperatures rose, the surface soil layer started to thaw. The dominant runoff component in this period was subsurface interflow (*Wang et al., 2017*). According to the previous investigation in the QTP grassland, SIC density at different soil depths showed a pattern of top 30 cm < top 50 cm < top 100 cm (*Yang et al., 2010a*). Thus, when the active layer gradually thawed to the deeper layer, more SIC was export by the

lateral flow. The higher the air temperature, the greater thickness of the active layer (*Wu & Zhang, 2010*), which could result in more soil interflow and deeper flow pathways. As soil thaw depth increased, the leaching potential of dissolved carbon from the soil also increased. Besides, the soil respired $CO_2$ also rise as active layer thaw (*Chen et al., 2017*). More soil $CO_2$ and soil water can form carbonic acid, which may enhance the mineral weathering and produce more DIC (*Zolkos, Tank & Kokelj, 2018*). Consequently, more DIC leached from the active layer to the water flow, and the DIC concentrations of river water increased.

Surprisingly, DOC concentrations were positively correlated with thaw depth (Fig. 6C). Study in a northern QTP stream found DOC concentrations negatively correlated with thaw depth due to less organic carbon source in the deeper soil layer (*Mu et al., 2017*). While in our study area, the SOC is 27.39, 33.15, and 28.69 mg/g for 0–5, 5–20, and 20–40 cm, respectively (*Chen et al., 2017*), which partly explained why the riverine DOC increased with thawed depth during spring thaw period. When the active layer thawed to the deeper soil layer where SOC is lower (*Ding et al., 2016*; *Mu et al., 2017*; *Yang et al., 2008*), the groundwater contribution rate increased as the thawing process proceeded (*Wang et al., 2017*). Therefore, the increased groundwater, which contains a high concentration of DOC (11.59 mg/L), overriding the potential effects of lower SOC supply in the deeper soil layer and OC decomposition (*Drake et al., 2015*; *Striegl et al., 2005*, *2007*). As a result, riverine DOC increased. Even so, whether such pattern is widespread across QTP permafrost rivers need further study. The dissolved carbon fluxes also increased with thawed depth as more river runoff and higher carbon concentrations during in deep thawed period.

During the freezing period, the active layer displayed a pattern of two-sided freezing (*Woo, 2012*). As the air temperature decreased, the surface layer began to freeze and cool downward. The frozen surface layer stopped the escape of heat from the middle layer. The bottom of the active layer also started to freeze at the same time. Therefore, the middle layer was still unfrozen during the early stage of the freezing process. Meanwhile, precipitation decreased rapidly during the freezing period. The frozen impermeable surface and bottom layer obstructed the infiltration of precipitation and groundwater to deeper layers (*Wang, Hu & Li, 2009*). As a result, the proportion of groundwater supply in river runoff increased when the frozen depth increased (*Wang et al., 2017*). The groundwater is characterized by high dissolved carbon concentrations (59.28 mg $L^{-1}$ of DIC and 11.59 mg $L^{-1}$ of DOC) than the river water and precipitation (Table 3). Under these circumstances, the DIC concentration and DOC concentration increase as the frozen depth increases. The decrease in dissolved carbon fluxes as the frozen depth increased was because of the rapidly decreased river discharge, which caused by reductions in precipitation and the gradual freezing of subsurface flow. Through the above analyses, we can see that active layer soil carbon exported to rivers was closely linked with thawing and freezing period as illustrated in the conceptual diagram (Fig. 8). Although our measurements of active layer thaw/freeze depths may have uncertainties, the pattern of active layer impacts still existed in 2 years of different meteorological and hydrological characteristics.

We consider the soil active layer freeze-thaw cycles to be the most important factors affecting the riverine dissolved carbon export in our site for the following four reasons.

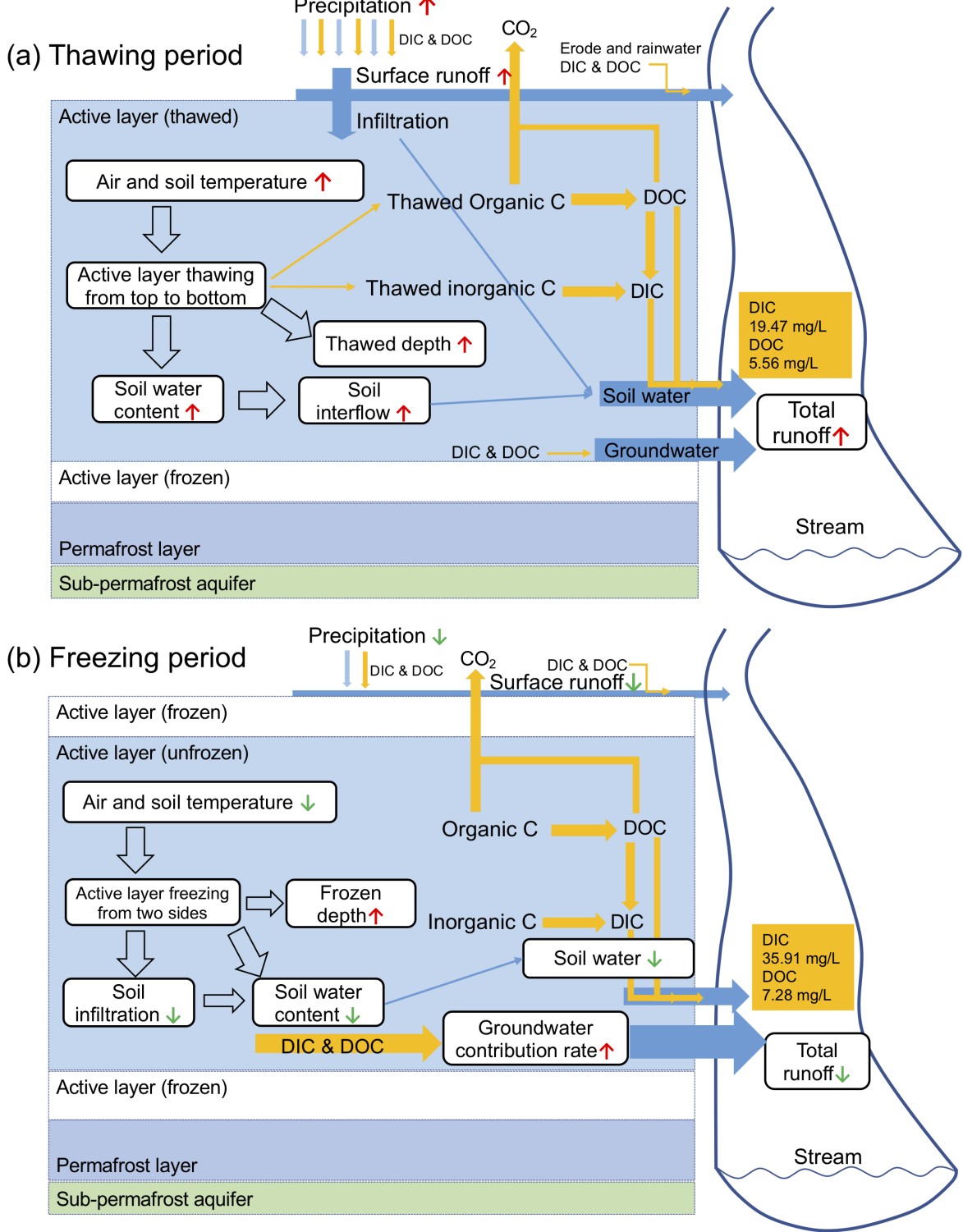

**Figure 8 Conceptual diagrams of riverine carbon export in thawing (A) and freezing (B) period.** Red upward arrows mean increase and green downward arrows mean decrease. Blue arrows mean water flow and yellow arrows mean carbon flow. Thawed depth is the thawed thickness of the active layer, while frozen depth is the sum thickness of frozen part in the figure.

First, the thawing and freezing processes controlled the thaw-released soil carbon from the active layer. As shown before, riverine dissolved carbon concentrations were closely related to active layer thaw and freeze depth. Second, permafrost river baseflow, which closely related to riverine carbon, was largely regulated by active layer thaw (*Smith et al., 2007*). Third, the widely distributed freeze-thaw erosion in the QTP, which can affect the lateral soil carbon export, is controlled by freeze-thaw cycles (*Zhang, Liu & Yang, 2007*; *Wang et al., 2017*). Fourth, the dissolved carbon fluxes were determined by discharge volumes, while surface runoff processes were primarily controlled by active layer freeze-thaw cycles rather than precipitation (*Wang, Hu & Li, 2009*). The systematic shifts of DIC and DOC we observed during the flow season mainly reflected the seasonal change of the active layer.

The active layer was predicted to become thicker in the future warming scenario (*Wu & Zhang, 2010*; *Yang et al., 2010b*; *Zhang, Frauenfeld & Serreze, 2005*). This prediction suggests earlier spring thaw and later autumn freeze up due to the increases in the length and depth of subsurface flow paths that will occur under climate warming (*Frampton et al., 2011*). The period during which lateral carbon losses occur in each year will be extended. Increased thaw depth could possibly extend the residence time of dissolved carbon in the flow path, which could affect the biodegradability of DOC (*Vonk et al., 2015b*). It then becomes inevitable that the active layer change will change the carbon export patterns. Additionally, changes in soil water content could affect the active layer thermal state and the thawing and freezing processes of the soil active layer (*Wang et al., 2012*), which may also accelerate the lateral carbon export. A warming climate and the corresponding increase in the thickness of the active layer (*Yang et al., 2010b*) may enhance this carbon loss through permafrost degradation and active layer thaw, which will eventually affect the regional carbon cycling.

## SUMMARY AND CONCLUSIONS

In this study, we showed that large amounts of terrigenous carbon were exported to rivers in headwater streams of the QTP continuous permafrost region. DIC is the overwhelmingly dominant form of dissolved carbon in our study watershed. The DIC flux constitutes more than 80% of the total dissolved carbon flux, with 3.95 g C m$^{-2}$ year$^{-1}$ of DIC and 0.94 g C m$^{-2}$ year$^{-1}$ of DOC fluxes. The higher DIC concentration than DOC may due to the higher SIC than SOC density. Dissolved carbon concentrations were closely related to baseflow, suggesting the importance of subsurface flow and groundwater supply to riverine carbon. Spatial distribution of dissolved carbon was mainly differed by land cover. Seasonal changes of DIC and DOC were mainly affected by freeze-thaw cycles. The river water cation data suggest carbonate weathering by soil $CO_2$ dissolution may also play a role in riverine DIC export. The active layer freeze-thaw cycles played important roles in the riverine dissolved carbon export of the QTP since freeze-thaw cycles controlled the thawed carbon release and catchment hydrological processes. Our findings provide new insights into the carbon cycling of this region and highlight the importance of soil active layer thawing and freezing processes on riverine carbon export in permafrost regions. Climate warming-induced permafrost degradation may result in higher disturbance of permafrost carbon and create a subsurface flow dominated hydrology system, which will

# PeerJ

lead to more lateral transport of dissolved carbon to river networks and change the regional carbon balance. More attention should be paid to the relationships between permafrost change and riverine carbon export.

## ACKNOWLEDGEMENTS

We thank the staffs of Fenghuo Mountain Observation Station of China Railway Northwest Institute for assisting our field work.

### Funding

This study was supported by the Major Research Plan of the National Natural Science Foundation of China (No. 91547203), the National Natural Science Foundation of China (No. 41890821, 41701037), and the Strategic Priority Research Program of Chinese Academy of Sciences (No.XDA20050102). Chunin Song also received support for this study from the China Scholarship Council. The funders had no role in study design, data collection and analysis, decision to publish, or preparation of the manuscript.

### Grant Disclosures

The following grant information was disclosed by the authors:
Major Research Plan of the National Natural Science Foundation of China: 91547203.
National Natural Science Foundation of China: 41890821, 41701037.
Strategic Priority Research Program of Chinese Academy of Sciences: XDA20050102.
China Scholarship Council.

### Competing Interests

The authors declare that they have no competing interests.

### Author Contributions

- Chunlin Song conceived and designed the experiments, performed the experiments, analyzed the data, contributed reagents/materials/analysis tools, prepared figures and/or tables, authored or reviewed drafts of the paper, approved the final draft.
- Genxu Wang conceived and designed the experiments, authored or reviewed drafts of the paper, approved the final draft.
- Tianxu Mao conceived and designed the experiments, performed the experiments.
- Xiaopeng Chen contributed reagents/materials/analysis tools.
- Kewei Huang contributed reagents/materials/analysis tools.
- Xiangyang Sun contributed reagents/materials/analysis tools.
- Zhaoyong Hu contributed reagents/materials/analysis tools.

### Data Availability

The raw measurements are available in the Supplemental File.

## Supplemental Information

Supplemental information for this article can be found online at http://dx.doi.org/10.7717/peerj.7146#supplemental-information.

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
