# Peer review of "Importance of active layer freeze-thaw cycles on the riverine dissolved carbon export on the Qinghai-Tibet Plateau permafrost region"

_PeerJ, doi:10.7717/peerj.7146_

## Round 0.1 · original submission · Minor Revisions

This MS provides some new findings of C in rives in Tibetan Plateau which is very sensitive to climate change. Before the MS can be accepted, several minor changes are needed. Authors should carefully compare your findings with published results, particularly the different results about DOC and DIC. A good explanation about the possible reasons for the differences in C content is needed. A thorough English editing is also needed.

[]

·

Basic reporting

See general comment

Experimental design

See general comment

Validity of the findings

No comment

Additional comments

Comment to“Importance of active layer freeze-thaw cycles on the riverine dissolved carbon export on the Qinghai-Tibet Plateau permafrost region”

In the context of global warming, the active layer and permafrost in the Qinghai-Tibet Plateau (QTP) are experiencing quick change, the severe permafrost degradation shows great effects on hydrological and biogeochemical processes, however, riverine dissolved carbon transport changes and their responses to freeze-thaw cycles remains poor. Authors used data in thaw and freeze seasons of 2014 and 2016, DIC and DOC concentrations and fluxes and their relationships with freeze-thaw cycles were consequently unveiled following a hypothesis. Generally, the Ms is interesting and provides good data of riverine C in the permafrost region. The structure, figure and Tables look good. While English should be improved (I have given some specific suggestions as follows). Methods were well described with sufficient detail while hypothesis should be well stated and should be more meaningful.

The authors get main findings of fluxes, concentrations and relative importance of DIC and DOC relating to thawing and freezing process. i.e., As the freezing process, DIC and DOC concentrations increased, while their fluxes decreased. As the thawing process, fluxes and concentrations of dissolved C increased. Authors discussed their patterns and controls, and also compared to other studies, while I think authors also should related to contrary patterns of other studies (i.e., L81-82 Thawing of active layer could result in the decrease of DOC and increase of DIC due to DOC mineralization and respiration (Drake et al., 2015; Striegl et al., 2005; 2007; L102-103 A previous study at northern QTP found that DOC export decreased as thaw depth increased (Mu et al., 2017). What resulted in the different results?.

Methods
Authors must clarify the measurements of thawed depth, frozen depth, land use composition etc.

Some specific comments including languages.
L 33 delete “while”
L43 change “was highly depended on” to “was highly dependent on”
L44 Please rewrite
L56 Please check
L67 Maybe change “Riverine exported carbon” “riverine carbon” or “riverine carbon export”?
L68 “according to studies conducted in a variety of scales and locations”, these words do not make sense, please delete
L79 “deepen”?
L85-87 Please re-organise
L106-107 Hypothesis testing is good, please be more clear
L127 What is the highest temperature? Please show the number
L136 Please Check the citation
L149 Your findings of major ions?
L155-157 I can not catch the sentence
L174 Please need to re-organize
L186 maybe “after sampling”
There are many typos such as the format of citations: L306, L313, L321, L327 any problem of citation? while I can not point out all issues

Tables
Table 1 Please indicate that how to get the catchment characteristics (i.e., land use composition)
Table 3 Please indicate the significant difference between DOC and DIC?

Figures
Fig S5 There are no plots in the ternary diagram?

Reviewer 2 ·

Basic reporting

When I am reviewing manuscripts, I usually raise several major comments and then give 2 or 3 pages of minor comments. But I read this manuscript, I found the field work was solid,and the manuscript was well organized, the data have been properly analyzed, and the conclusions were interesting. It is a pleasure to review this manuscript.

Experimental design

The experiments were designed rigorously.

Validity of the findings

It is interesting, and the results were in agreement with our expectations.

Additional comments

There are only minor suggestions for the authors, which I think they can do this work during the proof stage, so I need not see the revised version.

L76-77. Delete this sentence.
L77, The deepening AL could. You need not to repeat the seasonal thawed active layer.
L86, I would prefer passive voice.
L89, Seasonal change in soil thaw depth. The active layer is almost the same as the deepest thaw depth in the autumn.
L94, the largest. Not the only. Even in Hawaii, there is permafrost.
L94 wrong citation. It is Zou et al. 2017.
L97. There are several other reports, such as Zhao et al., 2018, Mu et al., 2015. Although Ding reported the SOC pools using SVM, there are also large uncertainties. For example, they used the kriging method to retrieve the spatial soil particle distribution, which is unacceptable.
L103, while it is unknown about the DIC exports changes during the freeze-thaw cycles in AL.
L107, delete potential.
L113-115. Remove these sentences.
L479, delete “since…freeze”.
L486, Climate warming-induced permafrost degradation may result in higher disturbance of permafrost carbon
L490 Connections-----relationships
L491. Delete this sentence.

·

Basic reporting

This is an interesting, solidly executed pieces of work. The topic is of relevance and the findings should be of interest to the international readership. The methodological approach is appropriate and is sufficiently described. The data are well presented and discussed, and the conclusions drawn are supported by the findings. The paper can be accepted after a thorough language editing.

Experimental design

The study design is appropriate.

Validity of the findings

The findings are justifiable.

Additional comments

This is an interesting, solidly executed pieces of work and I enjoyed reading it. The topic is of relevance and the findings should be of interest to the international readership. The methodological approach is appropriate and is sufficiently described. The data are well presented and discussed, and the conclusions drawn are supported by the findings. The paper needs, however, a thorough language editing before it can be published. Below are just a couple of examples of grammar/usage errors:

Line 30: Change “riverine exported carbon” to “riverine carbon export”

Line 32: Add “the” before “central QTP”

Lines 37-39: The sentence reads awkward. You may rephrase it to something like: “Spatially, DIC and DOC concentrations were positively correlated with vegetation coverage and were lowest in bare land.”

Line 45: Change “in QTP” to “in the QTP”

Line 192: Runkel et al 2004 is missing in Refs.

Figure 3 caption: Do you mean monthly average fluxes in kg per day? Please be clear.

Figure 3 caption: Is monthly mean discharge or actually total monthly runoff and precipitation?

---

## Round 0.2 · accepted · Accept

This MS has made contribution of our understnding of C cycle in Tibetan Plateau. In the revised version, you have made changes and your MS meets the publication standard.